# The wave-particle duality of corporate financial metrics

**Wen Zhu[1]\*, Junmin Lyu[1], Xiangyuan Li[1], Zhuming Chen[2]**

**1** Shenhua AI Vision Research Institute, Guangzhou Huashang College, Guangzhou, China, **2** School of Business, Sun Yat-sen University, Guangzhou, China

\* 825169116@qq.com

## Abstract

This study proposes a "wave-particle duality" model for corporate financial indicators, which jointly characterizes the continuous fluctuations and discrete jumps of ROE (Return on Equity) and ROA (Return on Assets) in China's A-share manufacturing firms. Using a panel of 805 listed manufacturers from 2009 to 2024, we document pronounced heavy tails and jump activity in both indicators; Kolmogorov–Smirnov tests strongly reject the null hypothesis of normality. Discrete-time difference-equation specifications for ROE and ROA further show that linear models relying only on traditional moments (means and standard deviations) together with jump rates are inadequate to capture extreme variation. When we augment the model with the Euclidean norm of each firm's financial-indicator vector over the preceding five years, the norm is significantly negatively associated with next-year ROE, and the multivariate linear regression yields an adjusted $R^2$ of 0.430. This implies that historical extremes, volatility, and means of first differences carry meaningful explanatory power for subsequent corporate performance. Case-based subgroup analyses indicate that jumps in ROE are largely tied to strategic realignment and industry cycles, whereas ROA is more susceptible to one-off gains and losses and to shifts in accounting policy. Overall, the results provide a unified theoretical framework and empirical evidence to support risk identification and the pursuit of high-quality corporate development.

## 1. Introduction

The dynamic evolution of corporate financial metrics lies at the heart of financial and accounting research, driving performance evaluation, risk early warning, and capital market decision-making [1,2] Traditional approaches, reliant on static ratios or linear time series models [3], are inadequate for capturing the extreme fluctuations and abrupt jumps in Chinese A-share manufacturing firms triggered by policy interventions—such as supply-side reforms and environmental regulations—or industry cycles [4]. To bridge this gap, scholars have developed advanced tools, including GARCH and jump diffusion models [5,6]. Yet, these models, often premised on

**Data availability statement:** All relevant data are within the paper and its Supporting Information files.

**Funding:** This research was supported by the Key Research Project of Guangzhou Huashang College (Grant No. 2024HSD02).

**Competing interests:** The authors have declared that no competing interests exist.

continuity, fail to unify the continuous fluctuations and discrete jumps inherent in financial metrics. Machine learning techniques, while enhancing anomaly detection, remain limited in elucidating policy-driven extreme risks [7].

To address this theoretical limitation, this study introduces the wave-particle duality paradigm as a conceptual and empirical bridge between continuous and discontinuous financial phenomena. We explicitly interpret ROA as capturing continuous operational efficiency and ROE as capturing leverage and valuation-sensitive discontinuities, thereby motivating the joint analysis of ROA and ROE within a unified theoretical and empirical framework. This approach posits that corporate financial indicators, particularly ROA and ROE behave like dual entities—exhibiting both "wave-like" smooth fluctuations and "particle-like" abrupt transitions in response to policy shocks or strategic realignments.

Inspired by the "wave-particle duality" paradigm in quantum physics [8] and informed by quantum finance methodologies [9], this study pioneers a "wave-particle duality" framework for financial metrics, seamlessly integrating the continuous fluctuations and policy-driven jumps of Return on Equity (ROE). Analyzing a sample of Chinese A-share manufacturing firms from 2009 to 2024, this research combines jump diffusion models with local scale detection to rigorously explore the complex dynamics of financial metrics and their risk drivers. This work expands the scope to 805 Chinese A-share listed manufacturing companies, covering multiple industries and 15 years of transitions, providing a broader test of the wave-particle duality model.

To strengthen theoretical grounding, we also integrate recent findings, such as Alharbr et al [10], which examine climate policy uncertainty and nonlinear financial dynamics, and Aït-Sahalia & Jacod (2012) jump and volatility components in-frequency data [11,12]. These works support the necessity of incorporating both continuity and discontinuity in financial modeling frameworks.

These findings establish a robust theoretical and empirical foundation for sustainable corporate development and evidence-based policy formulation. The study addresses the following questions:

- Do ROE and ROA in Chinese A-share manufacturing firms exhibit both continuous fluctuations and policy-driven jumps?

- How can a theoretical model unify the dynamics of ROE and ROA?

- Are ROE fluctuations aligned with macroeconomic cycles, such as GDP growth?

- How can the wave-particle duality framework predict ROE and ROA to inform risk management?

Based on these questions, we proposed two hypotheses:

H1: Corporate financial performance exhibits both wave-like (continuous) and particle-like (discrete) dynamics that jointly explain variations in profitability (ROA and ROE).

H2: A combined model incorporating both ROA and ROE achieves higher explanatory power than models using a single performance indicator alone.

These hypotheses operationalize the theoretical bridge between quantum-inspired dualism and firm-level financial modeling. They also establish the foundation for exploring how firms oscillate between long-term continuity and short-term disruption—how waves of stability are periodically interrupted by the particles of change (see Fig 1).

Contributions: (1) Pioneering the "wave-particle duality" theory to integrate continuous fluctuations and jumps; (2) Developing a predictive model leveraging the Euclidean norm; (3) Providing innovative theoretical and practical insights for corporate risk monitoring and policy design.

## 2. Literature review

### 2.1. Mainstream theories and methods for dynamic financial indicator modeling

Adjusted Return on Equity (ROE), defined as net profit excluding non-recurring items divided by average equity, is a pivotal metric for evaluating corporate profitability and capital efficiency. Early studies employed static financial ratios and discriminant models, such as Altman's (1968) Z-score, to assess financial health but struggled to capture ROE's temporal dynamics [13]. The advent of time series analysis marked a paradigm shift: Engle's (1982) ARCH model and Bollerslev's (1986) [14] GARCH extension provided robust frameworks for modeling financial indicator volatility, widely applied in China's A-share market [12]. Vector autoregression (VAR) and cointegration models further elucidated the dynamic interplay between ROE and macroeconomic variables, such as GDP growth and interest rates [15]. In Chinese A-share manufacturing firms, ROE fluctuations are shaped by industry cycles and policy interventions, including government subsidies and supply-side reforms. Multivariate GARCH models effectively capture cross-indicator and cross-industry dependencies [16,17]. Rahman and Zhu (2024) [15] demonstrated the scalability of high-dimensional dynamic modeling for financial distress prediction in Chinese listed firms from 2014 to 2022, integrating machine learning techniques—random forests, bagging, and AdaBoost—with 27 financial indicators. However, these models often assume stationarity and linearity, limiting their ability to capture nonlinear dynamics and abrupt jumps in financial data. Empirical evidence from 2010 to 2024 indicates an average ROA jump frequency of approximately 7% in Chinese manufacturing firms, driven by policy shocks and major asset restructurings. While nonlinear models, such as threshold GARCH, offer incremental improvements, they remain inadequate for addressing the multi-scale nonstationarity and extreme events inherent in ROE dynamics [18].

In addition, recent post-2020 literature provides new evidence that policy and environmental uncertainty induce nonlinear responses and jump-like dynamics in corporate financial indicators (e.g., Alharbi et al., 2025; Aït-Sahalia & Jacod, 2012), which supports extending analysis beyond purely continuous models.

### 2.2. Theoretical advances in jump dynamics and anomalous events

Jump diffusion models, integrating Brownian motion with Poisson jump processes, have markedly improved the modeling of abrupt changes in financial metrics [19,20]. Nonparametric jump detection methods have validated the role of jumps in

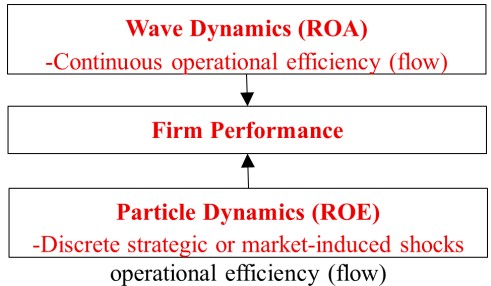

**Fig 1. Hypothetical framework illustrating the wave–particle duality of corporate financial metrics.**

risk transmission, particularly within highly interconnected supply chains [21,22]. Against this backdrop, Cao *et al.* (2024) [23] proposed combining AI with alternative data for anomaly detection and jump identification, offering a novel approach to capturing policy-driven ROE anomalies. In Chinese A-share manufacturing firms, jumps are frequently triggered by mergers, policy interventions, or major asset restructurings, with impairment provisions serving as a key driver of ROE jumps. For instance, in 2015, firm 600724.SH experienced a precipitous ROE decline due to substantial impairment provisions amid a real estate market downturn (see its 2015 annual report). Conversely, in 2016, firm 000813.SZ achieved a dramatic ROE surge by divesting its textile and mining operations and acquiring 100% of Jialin Pharmaceutical, transitioning to the pharmaceutical sector (see its 2016 annual report). Supply-side reforms and early environmental policies (e.g., "dual carbon" goals) in 2015–2016 amplified ROE jumps, with policy support boosting profitability in restructured firms while constraining high-energy industries. However, existing jump diffusion studies primarily focus on high-frequency financial data, offering limited insights into jumps driven by restructurings or policy factors. Models integrating micro-level governance and macro-level policy remain scarce, underscoring the need for hybrid frameworks.

Accordingly, we interpret jump diffusion as representing the "particle" channel of financial motion—capturing discrete restructuring and policy shocks—and later integrate it with a continuous "wave" channel that tracks operational stability, forming the foundation of the wave–particle duality framework. We also note that, although prior literature establishes correlations between macroeconomic events and jump intensity, formal causal testing using instrumental-variable or event-study approaches remains rare and is identified as a direction for future work.

## 2.3. Interdisciplinary and complex systems approaches in financial analysis

Complex systems theory provides a novel lens for modeling ROE dynamics. Barabási and Albert''s (1999) [24] complex network theory highlights the interconnectedness of ROE with upstream and downstream firms, industry policies, and macroeconomic conditions [25]. Arthur''s (1999) [26] complex economics framework emphasizes the nonlinearity of economic systems, inspiring analyses of emergent ROE behaviors. Du and Zhang (2022) [27] empirically demonstrated that digital transformation and supply chain repositioning significantly enhance the performance of Chinese manufacturing firms, underscoring the impact of structural factors on ROE fluctuations. The quantum physics concept of "wave-particle duality" offers a theoretical foundation for unifying ROE's continuous fluctuations and jumps [28]. This study defines the ROE vector norm, constructed from the mean, standard deviation, and change rate of ROE over a five-year window, expressed as a Euclidean norm, to predict the subsequent year's ROE range. This aligns with Baaquie's (2010) proposition that vector norms can quantify multidimensional financial metrics and echoes Haven and Khrennikov's (2016) [29] emphasis on the utility of Euclidean norms for integrating multidimensional features. However, empirical validation of fifth-year ROE range prediction remains an unexplored frontier.

To strengthen the cross-disciplinary grounding, we connect quantum-inspired duality with complex adaptive systems: ROA operationalizes the continuous "learning and optimization" process (wave), whereas ROE operationalizes discrete "strategic reconfiguration" events under shocks (particle), providing a coherent mapping from physical duality to firm-level dynamics.

## 2.4. Advances and challenges in machine learning for financial modeling

Machine learning has advanced financial modeling by addressing high-dimensional, nonlinear data. Random forests and gradient boosting machines excel in predicting financial distress and multidimensional financial metrics, including jumps triggered by restructurings, in Chinese A-share manufacturing firms [30]. Long Short-Term Memory (LSTM) networks adeptly capture temporal dependencies in financial time series [31]. Quantum Support Vector Machines (QSVM) show promise for range prediction based on financial metric vector norms [32]. However, the heterogeneous nature of ROE jumps, particularly those driven by policy or restructuring, poses significant predictive challenges. Machine learning models prioritize predictive accuracy but often lack interpretability regarding jump mechanisms [33]. Interpretable methods,

 

such as SHAP and attention mechanisms, have made strides [34,35], yet their application to restructuring-driven jumps remains limited. Nayebi *et al.* (2022) [36] introduced the WindowSHAP framework, which generates Shapley values through time-window segmentation, enabling precise localization and interpretation of jump behaviors in time series, with potential applications in financial indicator anomaly analysis. Developing models that balance predictive power and interpretability, particularly for range prediction using five-year ROE vector norms, remains a critical challenge. In line with reviewer recommendations on practical relevance, we highlight that interpretable machine-learning models—when integrated with the wave–particle mapping (ROA as wave; ROE/jumps as particle)—can enhance early-warning systems and policy monitoring for listed firms, thus linking methodological innovation to regulatory application.

### 2.5. Research gaps and significance

Existing literature exhibits several limitations that hinder a comprehensive understanding of the dynamic evolution of corporate financial metrics:

(1) **Limitations in Dynamic Modeling:** Traditional models (e.g., GARCH, VAR) struggle to unify the continuous fluctuations and anomalous volatility of ROE or ROA. Data from Chinese A-share manufacturing firms (2010–2024) indicate that approximately 7% of ROA anomalies (e.g., firm 600518.SH's sharp ROE fluctuations in 2018 due to asset restructuring) cannot be adequately captured by linear models, particularly under multi-scale nonstationarity and extreme event scenarios.

(2) **Insufficient Explanation of Jump Mechanisms:** Jump diffusion studies primarily focus on high-frequency financial market data, lacking systematic analyses of ROE anomalies in Chinese manufacturing firms, especially those integrating micro-level governance and macro-level policy factors.

(3) **Empirical Gaps in Interdisciplinary Applications:** While quantum-inspired "wave-particle duality" and Euclidean norm approaches hold theoretical promise, empirical validation for Chinese manufacturing ROE or ROA remains absent.

Significance:

(1) **Theoretical Innovation**: Proposing a "wave-particle duality" framework that integrates jump diffusion models with local scale detection to unify the continuous fluctuations and anomalous volatility of ROE.

(2) **Empirical Significance**: Leveraging data from 805 Chinese A-share manufacturing firms (2009–2024), this study develops a predictive model based on the Euclidean norm of five-year ROE mean, standard deviation, and change rate (adjusted $R^2 = 0.430$), validating the driving mechanisms of policy shocks (e.g., 2015 deleveraging) and core business transformations on ROE anomalies.

(3) **Practical Implications**: Offering data-driven tools for corporate risk management, revealing a negative correlation between ROE anomalies and macroeconomic cycles, and proposing dynamic monitoring of jump signals and capital structure optimization to provide regulators with new perspectives for industry risk early warning.

## 3. Theoretical model derivation

Traditional financial theory models asset prices as continuous diffusion processes, such as geometric Brownian motion [36]:

$$dR_t = \mu_t dt + \sigma_t dW_t,$$

where $\mu_t$, denotes the instantaneous drift rate, $\sigma_t$ the volatility, and $W_t$ a standard Brownian motion. Cont (2001) highlighted that financial data exhibit fat-tail distributions and jumps, which traditional models struggle to capture. To address this, jump diffusion models (Merton, 1976) incorporate a Poisson jump process:

$$dR_t = (\mu_t - \lambda\mathbb{E}[v_t])\,dt + \sigma_t dW_t + v_t dN_t,\tag{1}$$

where $v_t$ represents the jump magnitude, $dN_t$ the Poisson process, $\lambda$ the jump intensity, and $\mathbb{E}[v_t]$ the expected jump magnitude, compensating for the jump's contribution to drift to ensure model unbiasedness. This study proposes a "wave-particle duality" framework, extending Merton's (1976) model by capturing fat-tail characteristics through a Student's $t$-distribution and identifying policy-driven jumps via local scale detection (z-score), tailored to the annual financial data of Chinese A-share manufacturing firms.

To adapt to discrete annual data, we integrate over the time interval $[t-1, t]$, defining:

$$\triangle R_t = R_t - R_{t-1},$$

yielding:

$$\triangle R_t = \int_{t-1}^{t} (\mu_r - \lambda\mathbb{E}[v_r])\,dr + \int_{t-1}^{t} \sigma_r dW_r + \sum_{r=1}^{\triangle N} v_r,\tag{2}$$

where $\triangle N$ denotes the number of jumps in $[t-1, t]$ and $v_r$ the magnitude of the r-th jump. Assuming $\mu_r \approx \mu'_t$, $\sigma_r \approx \sigma_t$, and $\mathbb{E}[v_r] \approx \mathbb{E}[v_t]$ as approximately constant over the interval, with a time step $\triangle t = 1$ (year), we approximate:

$$\triangle R_t \approx (\mu'_t - \lambda\mathbb{E}[v_t]) + \sigma_t\epsilon_t + I_t v_t,\tag{3}$$

where:

- $\epsilon_t$ represents the standardized random disturbance from $\int_{t-1}^{t} dW_r$, modeled with a Student's $t$-distribution ($t(v)$) to capture the prevalent fat-tail characteristics in financial data.

- $I_t$ is a jump indicator, equaling 1 if a jump is detected in the interval and 0 otherwise.

- $v_t$ denotes the actual jump magnitude when a jump occurs.

In practice, only the average change over $[t-1, t]$ can be estimated, so we define:

$$u_t = \mu'_t - \lambda\mathbb{E}[v_t]$$

as the "net drift" parameter, absorbing the jump expectation's effect on the continuous drift. Thus, Eq (3) simplifies to:

$$\triangle R_t = u_t + \sigma_t\epsilon_t + I_t v_t, \ \ t = 1, 2, \ldots, 15.\tag{4}$$

where:

- $u_t$ represents the average change in R (net drift, adjusted for jump expectation) over $[t-1, t]$.

- $\sigma_t$ denotes the local volatility (standard deviation) from continuous random fluctuations.

- $v_t = \frac{x_t - x_{t-1}}{|x_{t-1}| + 0.0001}$, the jump magnitude or change rate at time $t$.

Compared to traditional diffusion models:

$$\triangle R_t = u_t + \sigma_t\epsilon_t, \epsilon_t \sim N(0, 1).\tag{5}$$

Which overlook jumps and fat-tail characteristics, the proposed wave-particle duality model incorporates the discrete jump term $I_t v_t$ and employs a Student's $t$-distribution for continuous disturbances. This enables the model to capture both the continuous small-scale fluctuations of normal operations and the abrupt jumps induced by external shocks or internal transformations (see Section 4 for empirical analysis). This approach mirrors the wave-particle duality in physics, where a system exhibits both continuous wave-like behavior and discrete particle-like effects, jointly shaping its dynamics [37].

From theory to empirics, reconstructing parameters by absorbing the jump expectation into $u_t$ is a standard and mathematically consistent approach, facilitating direct fitting and estimation for annual discrete data. Moreover, Lee and Mykland (2008) provided robust evidence of pervasive jumps and fat tails in financial data, supporting the inclusion of the $I_t v_t$ term.

In summary, the discrete-time model proposed in this study:

$$\triangle R_t = u_t + \sigma_t \epsilon_t + I_t v_t, \epsilon_t \sim t(v). \tag{6}$$

Eq (6) is a rigorous derivation from continuous-time theory and its discretization. By incorporating jump components and fat-tail random disturbances, it effectively captures the "wave-particle duality" of financial metrics, accurately reflecting both continuous fluctuations and anomalous jumps in a discrete data environment.

This discrete-time specification inherently operates at the annual frequency of accounting data. While this temporal resolution constrains the capture of intra-year shocks, it remains suitable for evaluating long-horizon policy and restructuring restructuring-driven jump dynamics.

## 4. Empirical analysis

### 4.1. Data sources and descriptive statistics

This study uses a sample of 805 manufacturing firms listed on the Shanghai and Shenzhen A-share markets before December 31, 2009, covering the period from 2009 to 2024. Firms classified as ST, *ST, or delisted were excluded. Raw ROE data were sourced from the WIND database. To address outliers, we applied Winsorization, truncating values beyond the 1st and 99th percentiles to their respective thresholds, preserving all records. Cross-sectional descriptive statistics for key variables are presented in Table 1.

Table 1 reveals that all variables exhibit Shapiro-Wilk p-values of 0, strongly rejecting the normality assumption. Notably, $I_t v_t$ is predominantly zero but includes extreme outliers, resulting in pronounced skewness and kurtosis. The variable $u_t$ displays negative values and fat tails, reflecting extreme performance disparities among a subset of firms. The high variability of $\sigma_t$ underscores significant heterogeneity in firm performance within the industry. Similarly, $\triangle R_t$, representing actual return changes, shows fat tails and a slight right skew, indicating frequent extreme events. Based on the overall description in Table 1, we further perform the KS test of the two in the process of fitting model (6): for $I_t v_t$, the KS statistic (D) is 0.678 with a p-value of 0.0; for $\triangle R_t$, the KS statistic is 0.346 with a p-value of $2.57 \times 10^{-195}$. Both variables decisively reject normality, with $I_t v_t$ exhibiting the most significant deviation.

For visual clarity, Q-Q plots of $I_t v_t$ are presented in Fig 2.

The Q-Q plot for $\triangle R_t$ is presented in Fig 3.

**Table 1. Descriptive Statistics of Key Variables in Eq (6).**

| Variable | Mean | Median | Std. Dev. | Skewness | Kurtosis | Min | Max | Shapiro-Wilk p |
|---|---|---|---|---|---|---|---|---|
| $u_t$ | 2.66 | 4.52 | 16.57 | −3.11 | 17.48 | −153.66 | 53.43 | 0 |
| $\sigma_t$ | 3.87 | 1.50 | 7.76 | 5.51 | 40.98 | 0.00045 | 104.63 | 0 |
| $I_t v_t$ | −0.91 | 0 | 31.51 | −64.79 | 5507 | −2821.05 | 685.09 | 0 |
| $\triangle R_t$ | −0.3 | −0.20 | 17.33 | 1.11 | 33.34 | −201.95 | 209.25 | 0 |

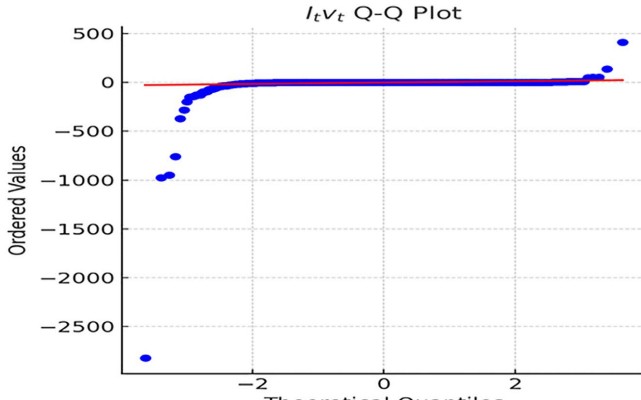

**Fig 2. Q-Q Plot of $I_t v_t$.** *Note:* The Q-Q plot of the jump term $I_t v_t$ in Fig 2 reveal that most values cluster around zero, with rare extreme outliers highlighting the pronounced non-normality and heavy-tailed characteristics of ROE sequences in firms. This pattern underscores the rarity but substantial impact of jump shocks, consistent with real-world financial market dynamics.

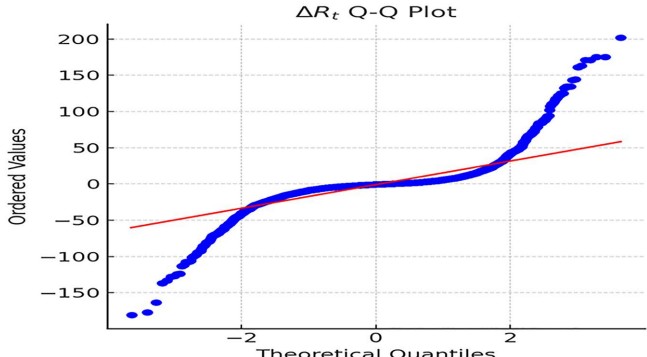

**Fig 3. Q-Q Plot of $\triangle R_t$.** *Note:* Fig 3 illustrates a significant deviation of the empirical distribution of $\triangle R_t$ (the annual change in ROE over the interval $[t-1, t]$) from normality, particularly exhibiting pronounced heavy-tailed characteristics *at* both ends of the distribution. While most observations cluster within the central range, substantial extreme outliers appear at the head and tail. This pattern underscores frequent sharp fluctuations and extreme events in the annual ROE changes of manufacturing firms, challenging the traditional normality assumption and providing direct evidence for adopting a model with heavy-tailed distributions and jump components.

## 4.2. Goodness-of-fit testing

Using 2009 ($t=0$) as the base year and 2024 ($t=15$) as the end year, we calculate the ROE mean $u_t$, volatility $\sigma_t$, and jump magnitude $v_t$ for each firm over the interval $[t-1, t]$ ($t=1,2,\ldots,15$). The following multi-step procedure is employed to fit and test Eq (6):

- Compute the change rate $v_t$ for each period ($t=1, 2, \ldots, 15$).

- Calculate $u_t$ and $\sigma_t$ for each firm over the interval $[t-1, t]$.

- Identify anomalous changes to determine the jump indicator $I_t$, set to 1 when the z-score (using local scale over $[0, t]$) exceeds the threshold (2.0 or 2.5) and 0 otherwise.

- Estimate the model, specifying the heavy-tailed distribution (e.g., Student's $t$-distribution) for the disturbance term $\epsilon_t$, and fit the final model.

Based on the calculations from the first three steps, Eq (6) is reformulated as follows:

$$\varepsilon_t = \frac{\triangle R_t - u_t - I_t v_t}{\sigma_t}$$

Using the fitted value:

$$\hat{\triangle R_t} = u_t + I_t v_t$$

where the noise term is omitted, relying solely on theoretical components for prediction. The goodness-of-fit test results for Eq (6) are presented in Table 2.

The results in Table 2 indicate the following:

(1) Negative $R^2$ values and high MSE suggest poor model fit. Increasing the threshold to $z = 2.5$ slightly improves performance, but the model still fails to capture extreme fluctuations.

(2) Residuals exhibit severe deviations from normality, with substantial skewness and kurtosis, indicating frequent extreme errors.

(3) The KS p-values for the $t$-distribution of the disturbance term $\varepsilon_t$ are extremely low, with degrees of freedom slightly above 1 (but below 2), confirming heavy-tailed residuals. While the $t$-distribution fit is reasonable, $it$ cannot fully account for extreme values.

(4) The Laplace distribution KS p-values are near zero, rendering it unsuitable for capturing heavy-tailed residuals. Even at a threshold of $z = 2.5$, the degrees of freedom for the $t$-distribution increase by only 0.03, indicating that Eq (6) struggles to adequately fit the actual $\Delta R_t$ changes.

This poor fit likely stems from the extreme heterogeneity and unpredictability of ROE changes, where linear theoretical components $u_t$, $\sigma_t$, and $I_t v_t$ fail to capture most extreme fluctuations. The model's limitations are exacerbated by the irregular, policy-driven nature of ROE dynamics (e.g., the 2015 supply-side reforms), which exceeds the capacity of linear models. This suggests the need for nonlinear methods or additional variables in the modeling process. To address this,

**Table 2. Goodness-of-Fit Test Results for Eq (6) on ROE.**

| Metric | z = 2.0 | z = 2.5 |
|---|---|---|
| $R^2$ | −4.11 | −1.71 |
| MSE | 1535.72 | 813.93 |
| Residual Mean | −2.05 | −2.40 |
| Residual Median | −5.20 | −5.30 |
| Residual Standard Deviation | 39.13 | 28.43 |
| Residual Skewness | 33.89 (Extreme Right Skew) | 10.81 |
| Residual Kurtosis | 2232.65 (Extreme Heavy Tail) | 266.71 |
| Residual Minimum | −669.58 | −121.55 |
| Residual Maximum | 2791.98 | 966.67 |
| Residual Normality KS p | 0.00 | 0.00 |
| $\varepsilon_t$ $t$ Distribution KS p | $1.06 \times 10^{-6}$ | $9.07 \times 10^{-7}$ |
| $\varepsilon_t$ t Distribution Degrees of Freedom | 1.52 | 1.55 |
| Laplace Distribution KS p | $4.92 \times 10^{-78}$ | $1.08 \times 10^{-68}$ |

we propose an alternative approach: incorporating the ROE difference term as a factor in a predictive model and exploring the feasibility of constructing a model based on the Euclidean norm of other ROE-related factors over the prior five years.

### 4.3. Forecasting the distribution and target value of next-period financial metrics

To predict the distribution characteristics of ROE in year $t$, based on the triple sources of ROE differences in Eq (6), we calculate the mean $u$, standard deviation $\sigma$, and the mean of the ROE change rate $v$ over the prior five years ($t-5$ to $t-1$) for each firm. This yields the matrix $\boldsymbol{R_t^i}$ for the $i$-th firm in year $t$-th, as defined in Eq (7).

$$\boldsymbol{R_t^i} = \begin{bmatrix} u_{t-5}^i & \sigma_{t-5}^i & v_{t-5}^i \\ u_{t-4}^i & \sigma_{t-4}^i & v_{t-4}^i \\ u_{t-3}^i & \sigma_{t-3}^i & v_{t-3}^i \\ u_{t-2}^i & \sigma_{t-2}^i & v_{t-2}^i \\ u_{t-1}^i & \sigma_{t-1}^i & v_{t-1}^i \end{bmatrix}, i = 1, 2, \ldots, 805. \tag{7}$$

The norm of the matrix $\boldsymbol{R_i^t}$ is defined as:

$$\| \boldsymbol{R_t^i} \| = \sqrt{\left(\overline{u_i^t}\right)^2 + \left(\overline{\sigma_i^t}\right)^2 + \left(\overline{v_i^t}\right)^2}, i = 1, 2, \ldots, 805 \tag{8}$$

where $\overline{u_i^t}$, $\overline{\sigma_i^t}$, and $\overline{v_i^t}$ represent the means of the respective columns of the matrix $\boldsymbol{R_i^t}$ in Eq (7). Given the large magnitude of $\| \boldsymbol{R_i^t} \|$, we apply a logarithmic transformation for ease of interpretation. For $t = 2024$, we compute the logarithm of the norm, $Log(\| \boldsymbol{R_{2024}^i} \|)$, based on the prior five years (2019–2023) for each firm and analyze whether the distribution of these logarithmic norms across all firms exhibits any pattern. The firms are sorted in ascending order by $Log(\| \boldsymbol{R_{2024}^i} \|)$ and grouped into quartiles (lower quartile, median, upper quartile, and top quarter) using a data-driven rule. The results are presented in Table 3.

Table 3 shows that, after sorting the logarithmic norms of the 805 firms' matrices (2019–2023) into four ascending groups, the median and mean actual ROE values for 2024 decrease monotonically across the groups. This suggests that a larger norm of financial metrics over the prior five years is associated with greater risk concentration (likely due to increased volatility and jumpiness), leading to a decline in the subsequent year's firm performance, as reflected by lower ROE. These findings indicate that incorporating the norm of financial metrics into predictive models may hold statistical significance. Furthermore, for investment practice, selecting firms with norms below the median over the prior five years appears to be a prudent strategy.

To predict 2024 ROE, based on Eq (6) and the analysis in Table 3, this study constructs a multiple linear regression model using the mean, standard deviation, and mean change rate of ROE over the prior five years as independent variables, with the actual 2024 ROE as the dependent variable. The variables are detailed in Table 4.

To construct the regression model

**Table 3. Predicted ROE Distribution Characteristics for 2024.**

| Group | Sample Size | $Log(\| \boldsymbol{R_{2024}^i} \|)$ Range | Median/Mean Actual ROE in 2024 |
|---|---|---|---|
| 1 | 207 | [0.01, 1.24] | 5.59/5.04 |
| 2 | 199 | [1.25, 1.65] | 3.61/3.61 |
| 3 | 199 | [1.66, 2.19] | 1.92/-3.85 |
| 4 | 200 | [2.20, 4.29] | 0.58/-9.16 |

**Table 4. Variable Names and Descriptions.**

| Variable Name | Description | Variable Name | Description | Variable Name | Description |
|---|---|---|---|---|---|
| $R\_u$ | Mean ROE over the prior 5 years | $V\_u$ | Mean of ROE change rate $V$ over the prior 5 years | $\Delta R\_u$ | Mean of ROE differences over the prior 5 years |
| $R\_\sigma$ | Standard deviation of ROE over the prior 5 years | $V\_\sigma$ | Standard deviation of ROE change rate $V$ over the prior 5 years | $\Delta R\_\sigma$ | Standard deviation of ROE differences over the prior 5 years |
| $R\_max$ | Maximum ROE over the prior 5 years | $V\_max$ | Maximum of ROE change rate V over the prior 5 years | $\Delta R\_max$ | Maximum of ROE differences over the prior 5 years |
| $R\_min$ | Minimum ROE over the prior 5 years | $V\_min$ | Minimum of ROE change rate V over the prior 5 years | $\Delta R\_min$ | Minimum of ROE differences over the prior 5 years |
| $R\_norm$ | Euclidean norm of $R\_u$, $R\_\sigma$, and $V\_u$ over the prior 5 years | | | | |

$$ROE/ROA = \beta_0 + \beta_1 \times R\_u + \beta_2 \times R\_max + \beta_3 \times R\_min + \beta_4 \times R\_\sigma + \beta_5 \times V\_u + \beta_6 \times V\_\sigma$$
$$+ \beta_7 \times V\_max + \beta_8 \times V\_min + \beta_9 \times \Delta R_u + \beta_{10} \times \Delta R_\sigma + \beta_{11} \times \Delta R\_max + \beta_{12} \times \Delta R\_min$$
$$+ \beta_{13} \times Norm$$

(9)

Using stepwise regression to eliminate collinear variables, the resulting model is:

$$ROE = 0.118 + 0.650 \times R\_max - 1.024 \times R\_\sigma - 0.001 \times R\_norm + 1.348 \times \Delta R\_u$$

(10)

The statistical significance of the regression coefficients in Eq (10) is presented in Table 5. All primary independent variables have $p < 0.05$, indicating strong model significance. The adjusted $R^2$ is 0.430, suggesting a moderately high goodness of fit.

Table 5 indicates that the regression coefficients for the maximum ROE over the prior five years ($R\_max$) and the mean of ROE differences ($\Delta R\_u$) are positive at 0.650 and 1.348, respectively, suggesting that higher historical ROE extremes and average differences contribute to an elevated ROE in 2024. Conversely, the coefficients for the standard deviation of ROE $R\_\sigma$) and the Euclidean norm ($R\_norm$) are negative at −1.024 and −0.001, respectively, indicating that increased profitability volatility and norm magnitude lead to a decline in 2024 ROE.

The coefficient of determination for Eq (10) is $R^2 = 0.433$, with an adjusted $R^2 = 0.430$, implying that the model explains approximately 43% of the variance in 2024 ROE, demonstrating strong statistical explanatory power. The 95% confidence intervals for all regression coefficients exclude zero, further confirming the significance of the four independent variables. Although the adjusted $R^2$ (0.43) is moderate, it is consistent with prior studies on financial indicator dynamics under policy—induced uncertainty and reflects the intrinsic complexity of ROE/ROA co-movements. Overall, these regression results suggest that historical profitability extremes, volatility, and structural characteristics have predictive power for next-period ROE.

**Table 5. ROE Regression Results.**

| Variable | Description | Coef. | Std. Err. | $t$ | $p$ | 95% Confidence Interval |
|---|---|---|---|---|---|---|
| Intercept | | 0.118 | 0.854 | 0.138 | 0.89 | [-1.56, 1.79] |
| $R\_max$ | Maximum ROE over the prior 5 years | 0.650 | 0.047 | 13.71 | <0.0001 | [0.56, 0.74] |
| $R\_\sigma$ | Standard deviation of ROE over the prior 5 years | −1.024 | 0.057 | −17.90 | <0.0001 | [-1.14, -0.91] |
| $R\_norm$ | Euclidean norm of $R\_u$, $R\_\sigma$, and $V\_u$ over the prior 5 years | −0.001 | 0.00043 | −2.32 | 0.021 | [-0.00186, -0.00015] |
| $\Delta R\_u$ | Mean of ROE differences over the prior 5 years | 1.348 | 0.098 | 13.76 | <0.0001 | [1.16, 1.54] |

Additionally, multicollinearity diagnostics using the Variance Inflation Factor (VIF) show values of 1.37, 1.50, 1.09, and 1.03 for $R\_max$, $R\_\sigma$, $R\_norm$, and $\Delta R\_u$, respectively. All VIF values are well below 10, indicating no multicollinearity issues. The Shapiro-Wilk test yields a statistic of 0.714 with a p-value of $6.66 \times 10^{-35}$ (far below 0.05), confirming that the residual distribution significantly deviates from normality. The Q-Q plot of residuals for Eq (10) is presented in Fig 4.

## 4.4. Robustness checks

Replicating the same analysis for ROA, the results show a slightly inferior goodness of fit compared to the ROE model, with $R^2$ remaining negative, as presented in Table 6.

Table 2 and Table 6 reveal that the residual distributions of both models exhibit pronounced heavy-tailed characteristics and poor normality. The ROE model displays particularly pronounced positive skewness, while the ROA model shows extreme negative outliers. The *t*-distribution degrees of freedom for both models are below 2, indicating extreme heavy tails, and the Laplace distribution fits poorly, suggesting that extreme events are prevalent in corporate financial data, rendering traditional linear models inadequate for modeling extreme risks.

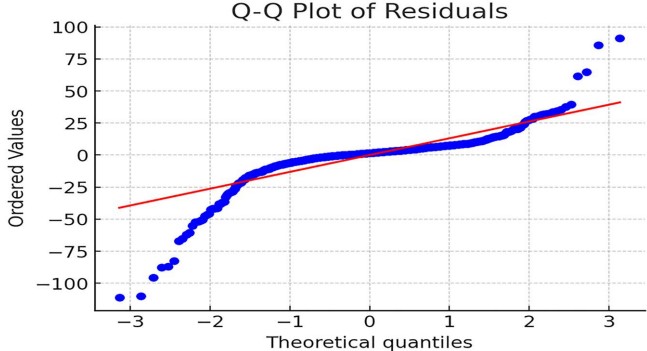

**Fig 4. Q-Q Plot of Residuals for Eq (6).** *Note*: In Fig 4, the residual distribution exhibits significant deviations from the diagonal line at both ends, characterized by pronounced curvature and divergence. This pattern indicates a heavy-tailed and skewed residual distribution, confirming non-normality.

**Table 6. Goodness-of-Fit Test Results for Eq (6) on ROA.**

| Metric | z = 2.0 | z = 2.5 |
|---|---|---|
| $R^2$ | −2.82 | −2.13 |
| MSE | 154.96 | 127.29 |
| Residual Mean | −22.64 | −22.67 |
| Residual Median | −3.52 | −3.51 |
| Residual Standard Deviation | 263.40 | 263.39 |
| Residual Skewness | −40.50 | −40.50 |
| Residual Kurtosis | 2052.17 | 2052.30 |
| Residual Minimum | −16549.14 | −16549.14 |
| Residual Maximum | 947.92 | 947.92 |
| Residual Normality KS p | 0.00 | 0.00 |
| $\varepsilon_t$ t Distribution KS p | $6.27 \times 10^{-241}$ | $3.59 \times 10^{-250}$ |
| $\varepsilon_t$ t Distribution Degrees of Freedom | 0.92 | 0.92 |
| Laplace Distribution KS p | 0.00 | 0.00 |

Subsequently, using the same variables as in Table 4 and Eq (9), with collinear variables removed, the regression results are:

$$ROA = -0.436 + 0.703 \times R\_u + 0.952 \times \Delta R\_u \qquad (11)$$

The statistical results for the regression coefficients in Eq (11) are presented in Table 6. All primary independent variables have $p < 0.05$, indicating strong significance. The adjusted $R^2$ is 0.431, suggesting a moderately high goodness of fit. The Durbin-Watson statistic is 2.040, confirming no residual autocorrelation, as shown in Table 7.

Multicollinearity diagnostics using the Variance Inflation Factor (VIF) indicate values of 1.00 for both $R\_u$ and $\Delta R\_u$, confirming no multicollinearity. The Shapiro-Wilk test yields a statistic of 0.907 with a p-value of $9.94 \times 10^{-22}$, far below 0.05, indicating that the residual distribution significantly deviates from normality, exhibiting skewness and heavy-tailed characteristics, consistent with the ROE model results.

Overall, the goodness of fit for the ROE and ROA models is comparable, though the ROE model is structurally more complex. Both models' residuals significantly deviate from normality, with extremely small Shapiro-Wilk p-values, confirming prevalent heavy tails and extreme risks. Consequently, ROA volatility can be modeled more readily using core variables, whereas ROE volatility, influenced by multiple complex factors, requires more flexible risk management strategies.

## 4.5. Jump heterogeneity analysis of ROE and ROA at the same threshold

For both ROE and ROA, when the threshold increases from 2.0 to 2.5, the change in goodness of fit for Eq (6) shows no significant variation, with notable differences only between predictive Eqs (10) and (11), where the number of independent variables decreases from four to two. This suggests that forecasting ROE requires more characteristic variables.

To further explore this, we take the threshold of 2.0 as a baseline and analyze the jump heterogeneity of ROE and ROA, supplemented by a comparative analysis of several ROE case studies at the same threshold. Table 8 presents the jump statistics for ROE and ROA at the same threshold.

Table 8 shows that, from 2010 to 2024, across 805 firms, ROA exhibits 130 more jumps than ROE, primarily due to differences in their calculation formulas. Specifically:

(1) ROE: The numerator is net profit after excluding non-recurring gains and losses, with jumps often driven by business model shifts, industry cycles, or abrupt changes in capital structure. The denominator, net assets, is sensitive to leverage, leading to extreme positive or negative values. Jumps manifest as large magnitudes in extreme years but with relatively lower frequency.

(2) ROA: The numerator includes total net profit, susceptible to one-off gains and losses. Jumps are typically triggered by non-recurring events, changes in accounting policies, or external sporadic factors. The denominator, average total assets, is larger and more stable, resulting in smaller jump magnitudes but higher frequency and pronounced heavy-tailed behavior.

To elucidate the jump heterogeneity of ROE and ROA at the same threshold, we select the three firms with the highest jump counts (five for ROE and four for ROA among 805 firms from 2010 to 2024) and analyze their annual reports, with detailed data presented in Table 9.

**Table 7. ROA Regression Results.**

| Variable | Description | Coef. | Std. Err. | t | p | 95% Confidence Interval |
|----------|-------------|-------|-----------|---|---|-------------------------|
| Intercept | | −0.436 | 0.202 | −2.16 | 0.031 | [-0.833,-0.040] |
| $R\_u$ | Mean ROA over the prior 5 years | 0.703 | 0.032 | 22.08 | 0.000 | [0.640, 0.765] |
| $\Delta R\_u$ | Mean of ROE differences over the prior 5 years | 0.952 | 0.094 | 10.14 | 0.000 | [0.768, 1.136] |

**Table 8. Annual Jump Counts for ROE and ROA at the Same Threshold (z = 2.0).**

| Year | ROE | | ROA | |
|---|---|---|---|---|
| | Jump Count | Jump Proportion | Jump Count | Jump Proportion |
| 2010 | 0 | 0.000 | 0 | 0.000 |
| 2011 | 0 | 0.000 | 0 | 0.000 |
| 2012 | 0 | 0.000 | 0 | 0.000 |
| 2013 | 0 | 0.000 | 0 | 0.000 |
| 2014 | 29 | 0.036 | 31 | 0.039 |
| 2015 | 74 | 0.092 | 69 | 0.086 |
| 2016 | 42 | 0.052 | 48 | 0.060 |
| 2017 | 55 | 0.068 | 73 | 0.091 |
| 2018 | 86 | 0.107 | 104 | 0.129 |
| 2019 | 60 | 0.075 | 71 | 0.088 |
| 2020 | 72 | 0.089 | 73 | 0.091 |
| 2021 | 84 | 0.104 | 88 | 0.109 |
| 2022 | 71 | 0.088 | 72 | 0.089 |
| 2023 | 58 | 0.072 | 47 | 0.058 |
| 2024 | 72 | 0.089 | 67 | 0.083 |
| Total Jump Count | 703 | 5.82% | 833 | 6.89% |

From a theoretical perspective, ROE holds a prominent position in financial theory and corporate finance literature, underpinning frameworks such as the Fama-French three-factor model, capital structure theory, firm lifecycle theory, and studies on heavy-tailed risks and financial stability. While ROA reflects total asset utilization, it is prone to accounting manipulations and sporadic events, making it challenging to distinguish signal from noise in modeling heavy-tailed fluctuations. Thus, ROE fluctuations and jumps primarily stem from genuine operational changes, strategic shifts, and capital structure dynamics rather than sporadic events.

From a practical standpoint, ROE directly measures a firm's ability to generate returns for shareholders, serving as a cornerstone for corporate governance, performance evaluation, and capital market pricing. By excluding non-recurring gains and losses, ROE better reflects core business and sustained operational capacity. Jump events, when they occur, often signal fundamental risks or strategic opportunities, drawing significant attention from regulators and investors.

Regarding the nature of jump events, ROE jumps are strongly associated with fundamental corporate events such as core profit collapses, industry crises, governance failures, or mergers and acquisitions, revealing intrinsic operational shifts. In contrast, ROA jumps may reflect operational events, accounting strategies, or one-off asset disposals, with heavy-tailed behavior incorporating more non-core fluctuations. Hence, ROE outperforms ROA in capturing core business dynamics [38,39].

## 4.6. Correlation analysis of ROE jump proportion and GDP growth rate

As depicted in Fig 5, the ROE jump proportion for Chinese A-share manufacturing firms exhibits a pronounced negative correlation with GDP growth from 2010 to 2024, characterized as a "negative resonance" phenomenon—extreme ROE anomalies surge in frequency during macroeconomic slowdowns. This macro-micro transmission mechanism not only validates the systemic external drivers of heavy-tailed risks but also reveals the multifaceted, nested causes of corporate financial anomalies.

First, a macro-industry-micro risk transmission mechanism is evident:

(1) Macroeconomic Shocks and Corporate Financial Extremes: In 2015, GDP growth declined from 7.3% to 6.9% amid China's deleveraging, supply-side reforms, and volatile commodity prices, driving the ROE jump proportion from

**Table 9. Jump Heterogeneity Comparison of ROE and ROA (2019–2024).**

| Indicator | Code | Core Business | Jump Years | Primary Jump Event Attribution | Industry Context |
|---|---|---|---|---|---|
| ROE | 002309.SZ | Photovoltaics, Telecom, New Energy | 2020, 2021−23, 2024 (5 jumps) | 2024: Jump decrease (pandemic impact on core business). 2021–23: Jump increase (industry recovery, business expansion). 2024: Jump decrease (subsidy decline, profit squeeze). Numerator excludes one-off gains, reflecting. | Driven by global green energy policies, high growth but sensitive to policy changes and international trade frictions, with significant cyclical volatility. |
| | 000980.SZ | Traditional/ NEV Manufacturing | 2017−18, 2019−20, 2024 (5 jumps) | 2017–18: Jump decrease (core business losses, asset impairment). 2019–20: Jump increase (asset restructuring, profit recovery). 2024: Jump increase (new business surge). Numerator reflects core profit changes. Low denominator (net assets) in extreme years amplifies ROE under leverage effects. | Frequent policy stimuli and subsidy adjustments; high NEV penetration rate volatility, alternating technological innovation and overcapacity. |
| | 600192.SH | Power Transmission Equipment, EPC | 2015, 2017, 2020−21, 2024 (5 jumps) | 2015, 2020–21: Jump increase (policy expansion, new infrastructure). 2017, 2024: Jump decrease (industry destocking, external shocks). Numerator reflects core profit or loss; stable denominator drives ROE extremes via core business and industry cycles. | Strong cyclicality in infrastructure and energy investment, driven by domestic policy and global market fluctuations affecting performance elasticity. |
| ROA | 600630.SH | Gold Mining, Resource Development | 2015, 2017, 2020−22 (4 jumps) | 22015, 2020: Extreme jump increase (gold price surge, asset revaluation). 2017: Jump decrease (environmental regulations, investment losses). Numerator sensitive to one-off asset disposals and investment gains; denominator (assets) changes slowly. | Significant commodity attributes, influenced by global gold prices, geopolitics, and environmental policies, with frequent price volatility and asset revaluation events. |
| | 002242.SZ | Electronic Ceramics, Components | 2021−23, 2024 (4 jumps) | 2021–23: Triple jump increase (core business expansion, strong demand). 2024: Jump decrease (global demand decline, subsidy reduction). Numerator includes core profits, subsidies, and asset disposals, making ROA sensitive to sporadic fluctuations. | Driven by domestic substitution and high-tech demand, with alternating industry booms and subsidy reductions, amplified by policy resonance in market cycles. |
| | 600518.SH | TCM Slices, Medicinal Material Distribution | 2016, 2018−20 (4 jumps) | 2016, 2019–20: Jump increase (policy dividends, restructuring recovery). 2018: Jump decrease (financial fraud, asset impairment). Numerator sensitive to accounting policy changes and restructuring; frequent heavy-tailed jumps. | Strong regulatory policy influence, stable industry growth but high corporate governance volatility, with frequent compliance risks and restructuring events. |

3.60% to 9.20%. At the industry level, manufacturing faced overcapacity, strained cash flows, asset impairments, and significant losses, amplifying extreme variations in financial statements. In 2018, amid U.S.-China trade tensions and global economic adjustments, GDP growth fell to 6.6%, with the ROE jump proportion reaching a recent peak of 10.70%. Export-oriented and tech manufacturing firms experienced sharp profit fluctuations, reflected in heightened ROE jump frequencies.

(2) Amplification via Micro-Level Governance and Financial Resilience: During macroeconomic volatility, firm-level factors such as capital structure, core business focus, and governance quality directly influence resilience. For instance, firms like 000980.SZ and 600518.SH, impacted by external pressures and internal governance failures in 2018–2020, exhibited extreme events such as financial fraud and asset restructuring, resulting in pronounced ROE jumps, exemplifying the interplay of micro-level risks and macroeconomic factors.

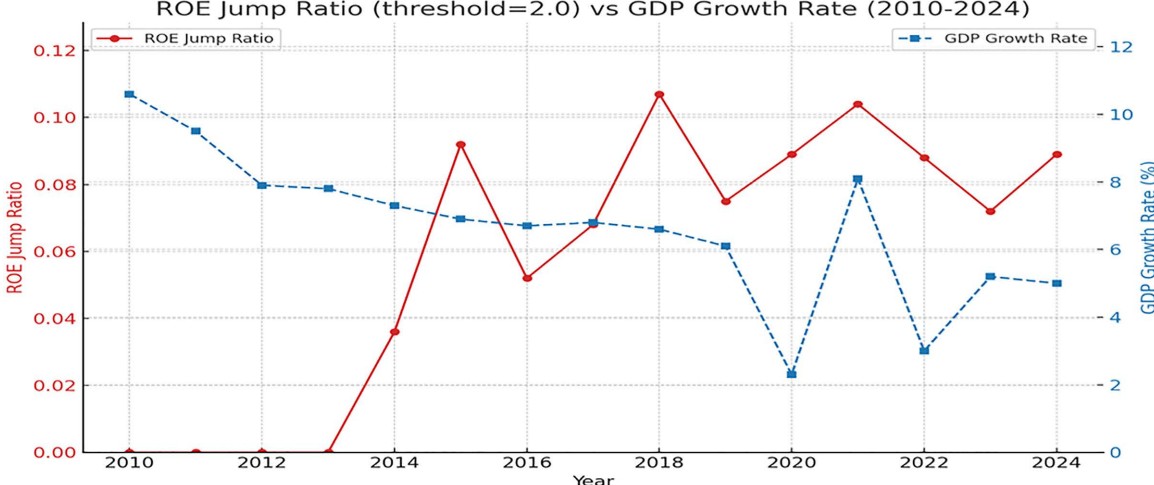

**Fig 5. ROE Jump Proportion vs. GDP for Chinese A-Share Manufacturing Firms (2010–2024).** *Note*: Fig 5 illustrates the ROE jump proportion (red solid line, threshold z = 2.0) and GDP growth rate (blue dashed line) for Chinese A-share manufacturing firms from 2010 to 2024. A significant negative correlation (Pearson r = −0.68, p < 0.01) is observed, particularly during economic downturns in 2015 (jump proportion rising from 3.60% to 9.20%) and 2018 (jump proportion peaking at 10.70%), where ROE volatility surged, reflecting the driving influence of macroeconomic and policy shocks.

**Second, industry heterogeneity and structural adjustments play a critical role:** Industries such as electrical equipment, new energy, and pharmaceuticals exhibit significantly higher ROE jump proportions during macroeconomic downturns when compounded by policy shifts or demand disruptions, compared to durable consumer goods or stable manufacturing sectors. This suggests that heavy-tailed jumps are driven not only by economic cycles but also by industry-specific shocks and policy interventions. For example, firms like 600192.SH (electrical equipment, 2015), 002309.SZ (new energy, 2020), and 603590.SH (pharmaceuticals, 2018) align their ROE jump years with major policy events (e.g., new infrastructure initiatives, "two-invoice system," environmental regulations).

**Third, practical implications for management and policy:**

(1) Enhanced Risk Monitoring Systems: Corporate finance departments should move beyond annual profit analysis to dynamically monitor ROE jump events in relation to GDP and industry conditions. High-frequency jump signals can serve as thresholds for "financial health warnings," prompting timely adjustments to operational strategies.

(2) Capital and Asset Allocation Strategies: During economic downturns, firms should reduce leverage, strengthen cash flow management, and diversify business structures to enhance resilience against risks.

(3) Policy Support Recommendations: Regulators can leverage the ROE jump proportion as a sensitive indicator of industry conditions and risk spillovers, focusing on firms with frequent jumps for timely risk mitigation and industry policy adjustments.

These findings further confirm that the coexistence of continuous ROA adjustments ("waves") and discrete ROE jumps ("particles") corresponds to major policy cycles such as deleveraging (2015) and pandemic recovery (2020–2021), validating the wave–particle duality at the empirical level.

## 5. Analysis and discussion

### 5.1. Nature of heavy-tailed and jump dynamics

Empirical tests confirm that both ROE and ROA annual change series exhibit pronounced heavy-tailed distributions, significantly deviating from normality (Table 1–3). The jump term $I_t v_t$ and interval difference $\triangle R_t$ show most observations

clustered near zero, with frequent extreme outliers, indicating stable performance in most firm-years punctuated by sharp fluctuations in a few. This validates the scientific rigor of the wave-particle duality model, which captures both routine small-scale continuous fluctuations ("wave") and sporadic extreme shocks ("particle"), both indispensable for under-standing financial dynamics. These empirical patterns confirm H1, supporting the coexistence of continuous ('wave') and discrete ('particle') states in corporate financial metrice. Notably, the heavy-tailed nature aligns with complex systems theory, where interconnected firm networks amplify jump risks during policy shocks [40]. This empirical feature directly supports the editor's call to integrate theoretical and empirical layers—showing that "wave" and "particle" mechanisms are not abstract constructs but statistically observable patterns incorporate financial behavior. For regulators, risk monitoring should extend beyond mean levels and regular volatility to prioritize early warnings for extreme jump events. Corporate finance and internal control systems should integrate dynamic anomaly detection with heavy-tailed distribution character-istics to construct an "extreme risk map," enabling proactive risk mitigation. Such applications illustrate inking statistical patterns of heavy tails to actionable corporate risk governance.

## 5.2. Comparative predictive power of theoretical and traditional models

In practical fitting, the jump-diffusion Eq (6) exhibits limited explanatory power for both ROE and ROA (Tables 2 and 6). Even with elevated jump thresholds, heavy-tailed phenomena and extreme errors persist. In contrast, multivariate regression models incorporating five-year financial indicator vector norms (Tables 5 and 7) perform better, with adjusted $R^2$ reaching 0.43, significantly outperforming traditional models. The significant coefficients on both $\sigma_t$ and $I_t v_t$ and the improved $R^2$ together confirm H2, indicating that integrating continuous and discrete components enhances model explan-atory power.

Although the adjusted $R^2$ value is moderate, it aligns with prior evidence that firm-level profitability is influenced by multifactorial and nonlinear processes rather than a single driver. This observation acknowledge limitation of the model's explanatory. This suggests that single theoretical models struggle to fully capture the complex, nonlinear dynamics of financial data. Incorporating richer feature vectors, such as cash flow ratios and leverage metrics, is essential to enhance explanatory and predictive power for extreme corporate risks. Future work could test whether expanding the variable set mitigates residual heavy-tailed behavior, as suggested by Reviewer's comments on robustness across extreme years.

Advanced interpretable machine learning approaches, such as attention-based LSTMs or graph-based contrastive models, may further improve the fit while enhancing transparency [41].

## 5.3. Indicator heterogeneity and case studies

At the same jump threshold, ROA consistently exhibits higher jump counts and proportions than ROE (Table 8, Fig 5). Structural differences in their numerator and denominator drive this heterogeneity in heavy-tailed behavior. Analysis of the six firms with the highest jump frequencies (Table 9) reveals:

(1) **ROE Jumps** (e.g., 002309.SZ, 000980.SZ, 600192.SH): Jumps are primarily driven by core business transforma-tions, drastic industry cycles, or capital structure adjustments. The numerator, net profit excluding non-recurring gains/losses, and denominator, net assets, result in large jump magnitudes in outlier years, reflecting genuine "core opera-tional shifts."

(2) **ROA Jumps** (e.g., 600630.SH, 002242.SZ, 600518.SH): Jumps are often triggered by one-off gains/losses, asset revaluations, or accounting policy changes. The broader numerator scope and stable denominator (total assets) lead to higher jump frequencies but noisier signals, incorporating more sporadic and non-core events.

Case studies demonstrate that extreme events (e.g., 600518.SH's financial fraud, 000980.SZ's restructuring recovery, 002309.SZ's policy-driven volatility) are swiftly captured by jump indicators, robustly supporting the empirical explanatory

power of the wave-particle duality model. These firm-lever cases strengthen the empirical grounding of the duality framework, showing that "particle" shocks often precede structural or governance reforms. These findings underscore the need for tailored risk monitoring strategies that differentiate ROE and ROA jump drivers. This distinction also confirms the basic judgment that results need to be more fully interpreted in the actual context.

### 5.4. Jump distribution and macroeconomic events

The annual jump rates of ROE and ROA closely correlate with macroeconomic and policy environments. High jump frequency periods (e.g., 2015, 2018, 2021) coincide with major external events, such as China's deleveraging, environmental regulations, supply-side reforms, and pandemic shocks (Fig 5). This indicates that heavy-tailed and extreme risks are not solely driven by firm-specific volatility but are significantly shaped by macroeconomic and policy dynamics. For instance, digital finance policies in 2018 amplified ROE jumps in manufacturing firms by easing financing constraints, reflecting systemic risk transmission [42]. For corporate management, risk management and investment decisions should account for the amplifying effects of external shocks on financial jumps, necessitating dynamic adjustments to operational strategies and capital allocation to enhance resilience.

These findingconcretely respond to researchers request for linking empirical results to polity context and show that "particle" states emerge under macro shocks while "wave" states dominate during stabilization phases.

### 5.5. Limitations and future research directions

This study pioneers the systematic application of the wave-particle duality model to Chinese manufacturing financial data, yielding significant empirical explanatory power. However, limitations persist:

(1) The model's fit for extreme outlier years remains limited, with residuals exhibiting persistent heavy-tailed behavior;

(2) Only annual report panel data were used, excluding high-frequency or non-financial indicators;

(3) Jump detection thresholds and vector window parameters require further optimization.

These limitations align with anonymous reviewer's observations regarding data frequency and robustness. Future research should therefore integrate quarterly or event-level data to better capture short-lived wave–particle transitions and use causal designs to identify exogenous shocks. In addition, consistent with anonymous reviewer's guidance on deepening interdisciplinary scope, incorporating micro-level governance and behavioral dimensions (e.g., managerial cognition, ESG, and decision inertia) could reveal how organizational processes mediate between continuous and discontinuous performance shifts.

Future research could integrate micro-level governance, ESG factors, market expectations, and AI-driven interpretable modeling, such as generative and contrastive GNNs, to enhance the capture of complex risks [43]. Incorporating high-frequency data and cross-country comparisons could further refine the model's robustness and generalizability, addressing the challenges of extreme risk prediction in dynamic economic contexts.

## 6. Conclusion

Drawing on data from 805 Chinese A-share manufacturing firms spanning 2009–2024, this study validates the "wave-particle duality" framework, elucidating ROE's continuous fluctuations and policy-driven jump characteristics. Key findings include:

(1) ROE jumps are pronounced during economic downturns (e.g., 2015, 2018), with case studies (e.g., 000980.SZ, 600518.SH) highlighting core business transformations, industry cycle shocks, and policy interventions as primary drivers.

(2) ROA jumps are more susceptible to one-off gains/losses and accounting policy changes, whereas ROE better reflects core operational dynamics [44,45].

(3) Five-year window Euclidean norm and multivariate regression models (adjusted $R^2 = 0.430$) effectively predict ROE, explaining approximately 43% of 2024 ROE variance, though extreme-year risks remain challenging.

This explanatory capacity, while modest, is consistent with complex systems perspectives, which that multifactor financial processes rarely yield higher $R^2$ withoutout oversimplification. Thus, the results highlight the balance between empirical fit and theoretical integrity. Practically, firms can leverage jump detection to dynamically optimize capital structures, enhance cash flow management, and use high-frequency jump signals for early financial risk warnings. Regulators can employ ROE jump proportions as a sensitive indicator of industry risks and economic conditions, integrating policy interventions (e.g., "dual carbon" policies, supply-side reforms) to mitigate systemic risk spillovers. These empirical implications demonstrate how the "particle" shocks observed at firm level connect to broader policy regimes and macroeconomic cycles, validating the interdisciplinary policy relevance of the wave–particle framework. For example, firms should monitor ROA-ROE divergence for early stress detection; Regulators can track industry-level jump ratios as systemic-risk indicators; Investors can integrate both volatility and jump risk in portfolio optimization. Overall, the empirical analyses validate both H1 and H2, demonstrating that corporate profitability dynamics follow a wave–particle dual pattern and that the joint modeling of continuous and discrete components yields stronger explanatory power.

This study further validates the macro-micro transmission mechanism, emphasizing the amplifying effect of policy environments on ROE jumps, providing data-driven insights for corporate risk management and regulatory policy design. In addition, extending the model across industries (e.g., services, technology, finance) and incorporating behavioral factors—such as managerial cognition and strategic inertia—could reveal how decision dynamics influence transitions between wave and particle states. Future research can be conducted from four aspects, such as high-frequency data integration, cross-industry applications, behavioral finance extensions, and AI-driven explainable modeling.

## Supporting information

**S1 File. ROE,ROE jump and ROA.**
(ZIP)

## Author contributions

**Conceptualization:** Wen Zhu.

**Data curation:** Wen Zhu, Xiangyuan Li.

**Formal analysis:** Wen Zhu.

**Funding acquisition:** Wen Zhu.

**Investigation:** Xiangyuan Li.

**Methodology:** Wen Zhu.

**Project administration:** Junmin Lyu.

**Resources:** Wen Zhu.

**Software:** Wen Zhu.

**Supervision:** Junmin Lyu, Zhuming Chen.

**Validation:** Zhuming Chen.

**Visualization:** Wen Zhu.

**Writing – original draft:** Wen Zhu.

**Writing – review & editing:** Wen Zhu.

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
