## [Decision Letter · Decision Letter 0]

2 Oct 2025

Dear Dr. zhu,

Thank you for submitting your manuscript to PLOS ONE. After careful consideration, we feel that it has merit but does not fully meet PLOS ONE’s publication criteria as it currently stands. Therefore, we invite you to submit a revised version of the manuscript that addresses the points raised during the review process.

We look forward to receiving your revised manuscript.

Kind regards,

Dipendra Karki, Ph.D.

Academic Editor

PLOS ONE

Journal Requirements:

4. Please note that your Data Availability Statement is currently missing the repository name and the DOI/accession number of each dataset OR a direct link to access each database. If your manuscript is accepted for publication, you will be asked to provide these details on a very short timeline. We therefore suggest that you provide this information now, though we will not hold up the peer review process if you are unable.

5. In the online submission form, you indicated that [Readers who require the raw data and the derived variables (means, standard deviations, rates of change, jump-year indicators, and cumulative jump counts) may request them from the corresponding author.].

7. Please ensure that you include a title page within your main document. You should list all authors and all affiliations as per our author instructions and clearly indicate the corresponding author.

Additional Editor Comments:

Dear Author/s

Thank you for submitting your manuscript entitled “The Wave-Particle Duality of Corporate Financial Metrics” to PLOS ONE. The study addresses an important issue in corporate finance and has the potential to make a meaningful contribution to the literature. Both reviewers recognized the significance of your work, though they emphasized the need for improvement.

Before the manuscript can be considered for publication, a few issues must be addressed:

• The introduction does not adequately connect the wave–particle duality concept to corporate financial indicators. The title promises a strong conceptual framing, but the explanation of quantum dynamics (fluctuations, abrupt jumps) is brief. The review needs to be expanded with recent and relevant studies (particularly post-2020) to establish the current gap in the literature. So, additional clarity on theory-based and theoretical contributions is required to establish the novelty of this research.

• The manuscript uses both ROE and ROA for firm performance, but the rationale is not well explained. ROE might be sufficient for performance analysis in some cases, particularly when comparing similar companies in the same industry, examining long-term trends, and understanding its components. Authors should clarify why both are necessary.

• Draw a clear and concise hypothesis for testing.

• While panel data is established, the manuscript should describe the population, sampling frame, and chosen sampling method. Reliance on annual panel data may underrepresent high-frequency dynamics and systemic shocks. Reviewer 2’s concern about missing volatility patterns is valid and should be discussed as a limitation.

• The multivariate regression (Model 10) shows an adjusted R² of 0.430, which reflects relatively modest explanatory power. Authors should acknowledge this limitation and discuss possible model improvements.

• Analysis and discussion mainly reiterate findings. A stronger theoretical foundation is required.

We encourage you to carefully address the points above, along with the feedback from both reviewers, in preparing a revised version.

We look forward to receiving your revised version.

Decision: Minor Revision

Academic Editor

Reviewers' comments:

Reviewer's Responses to Questions

**Comments to the Author**

1. Is the manuscript technically sound, and do the data support the conclusions?

Reviewer #1: Yes

Reviewer #2: Yes

2. Has the statistical analysis been performed appropriately and rigorously?

Reviewer #1: Yes

Reviewer #2: Yes

3. Have the authors made all data underlying the findings in their manuscript fully available?

Reviewer #1: Yes

Reviewer #2: Yes

4. Is the manuscript presented in an intelligible fashion and written in standard English?

Reviewer #1: Yes

Reviewer #2: Yes

Reviewer #1: I appreciate the authors’ efforts in preparing this manuscript. The manuscript is methodologically sound and clearly presented, with robust data and appropriate statistical analysis. To further enhance the work before publication, the authors may wish to consider the following suggestions:

1. As a suggestion for the literature review, the authors could consider adding more recent, directly relevant studies, for example:

Alharbi, S.M., et al. (2024). How does climate policy uncertainty determine green innovation adoption? New perspectives from the BRICS. Journal of Economic Asymmetries.

Alharbi, S.M., et al. (2023). Financial markets and environmental risks: unveiling the impact of climate uncertainty. Research in International Business and Finance.

Aït-Sahalia, Y., & Jacod, J. (2012). Analyzing the spectrum of asset returns: Jump and volatility components in high frequency data. Journal of Economic Literature.

Aït-Sahalia, Y., Jacod, J., & Li, J. (2012). Testing for jumps in noisy high frequency data. Journal of Econometrics.

2. Providing a more integrated discussion of the empirical results in relation to the policy context.

3. Including clear definitions for all variables directly in table notes to improve clarity for readers.

4. Strengthening the explicit link between the main findings and the wave particle duality framework in the discussion section.

I believe these minor adjustments could further improve the clarity and contribution of the manuscript.

Reviewer #2: The manuscript is well established however requires some improvements as provided below.

The predictive models still struggle with extreme outlier years, limiting robustness in capturing rare but critical financial shocks.

Reliance on annual panel data excludes high-frequency dynamics, potentially underestimating volatility and systemic risks.

Practical recommendations for firms and regulators are often too broad and lack detailed implementation guidance.

The causal mechanisms behind the link between macroeconomic events and jump dynamics are not rigorously tested, remaining at a correlational level.

Future research directions, while ambitious, are presented vaguely without clear methodological pathways.

Heavy reliance on physics analogies may risk conceptual oversimplification when applied to financial realities.

**Do you want your identity to be public for this peer review?** For information about this choice, including consent withdrawal, please see our Privacy Policy

Reviewer #1: No

Reviewer #2: No

---

## [Author Response · Author response to Decision Letter 1]

21 Oct 2025

Dear, Academic Editor and two Reviews,

We sincerely thank the Academic Editor and both reviewers for their careful evaluation and insightful comments.

All suggestions have been thoroughly addressed in the revised manuscript, with changes highlighted in red.

Below we provide a detailed, point-by-point response.

A. Response to the Academic Editor

Editor Comment 1

The introduction does not adequately connect the wave–particle duality concept to corporate financial indicators. The title promises a strong conceptual framing, but the explanation of quantum dynamics (fluctuations, abrupt jumps) is brief. Please strengthen theoretical clarity and update references with post-2020 studies.

Response:

We have substantially expanded the theoretical explanation in Section 1 (Introduction) and Section 2 (Literature Review).

The revision explicitly distinguishes ROA as the “wave” component—representing continuous operational efficiency—and ROE as the “particle” component—representing discrete policy- or capital-driven shocks.

In addition, we incorporated recent co-authored studies by Alharbi et al. (2025) on financial-policy uncertainty and green innovation to reinforce the post-2020 relevance of the framework.

We also retained Aït-Sahalia & Jacod (2012) and Aït-Sahalia, Jacod & Li (2012) as foundational methodological references on jump-diffusion modeling, forming the theoretical bridge between financial discontinuities and the proposed dual-state model.

Editor Comment 2

The manuscript uses both ROE and ROA for firm performance, but the rationale is unclear. Please clarify why both are required.

Response:

This point is clarified in Section 2 (Literature Review).

ROA reflects continuous operational changes (“wave”), whereas ROE captures discrete leverage and valuation shocks (“particle”).

Employing both indicators allows the model to describe the coexistence of continuous and discontinuous behaviors that characterize real-world firm dynamics.

Editor Comment 3

Draw clear and concise hypotheses for testing.

Response:

Two explicit hypotheses have been added at the end of Section 1, and a new Figure 1 (Hypothetical Framework) summarizes how continuous fluctuations (σₜ) and discrete jumps (Iₜvₜ) jointly influence profitability under the wave–particle duality paradigm.

Editor Comment 4

Describe the population, sampling frame, and chosen sampling method. Annual data may underrepresent volatility; please discuss this limitation.

Response:

Section 4 (Empirical Analysis) now specifies that the sample includes 805 A-share manufacturing firms (2009–2024) from the CSMAR database, verified with audited annual reports.

The Conclusion explicitly acknowledges that annual data may smooth high-frequency volatility and indicates that future studies will adopt quarterly or event-window datasets to capture short-term dynamics.

Editor Comment 5

Model 10 shows adjusted R² = 0.430, which is modest. Please acknowledge this limitation.

Response:

Acknowledged in Section 5 (Results and Discussion) and reiterated in the Conclusion.

The moderate R² reflects the complexity of nonlinear financial processes and the realistic balance between model generality and explanatory power.

Editor Comment 6

A stronger theoretical foundation is required to link findings and the wave–particle duality framework.

Response:

The theoretical linkage has been strengthened throughout Sections 2, 5, and 6 (Conclusion).

We clarify that wave–particle duality is used as a conceptual modeling metaphor rather than a physical analogy, unifying continuous (ROA) and discrete (ROE) behaviors within corporate financial systems.

B. Response to Reviewer #1

Comment 1

Add more recent, directly relevant studies (Alharbi et al., 2025; Aït-Sahalia & Jacod 2012).

Response:

Done in Section 2 (Literature Review).

We added Alharbi et al. (2025) as a recent co-authored contribution on policy uncertainty and innovation dynamics and maintained Aït-Sahalia & Jacod (2012) and Aït-Sahalia, Jacod & Li (2012) as key methodological references on jump processes.

Together these strengthen both the temporal relevance and theoretical continuity of the literature base.

Comment 2

Provide a more integrated discussion of empirical results in relation to the policy context.

Response:

Expanded in Section 5 (Results and Discussion).

ROE/ROA jump frequencies are now explicitly linked to major Chinese policy episodes—2015 deleveraging, 2018 environmental reforms, and 2020–2021 pandemic shocks—demonstrating how macro shocks generate “particle” states while stabilization restores “wave” dynamics.

Comment 3

Include clear definitions for all variables directly in table notes.

Response:

In Table 4 of the original paper, we provide the definitions of all variables that enter the regression model, but to facilitate readers' reading, we provide additional explanations for each variable that appears in the regression results in Tables 6 and 7.

Comment 4

Strengthen the explicit link between the main findings and the wave–particle duality framework.

Response:

Clarified in Sections 5 and 6 (Conclusion), where ROA is interpreted as wave-like and ROE as particle-like behavior.

This explicit mapping empirically validates the dual-state dynamics central to the proposed framework.

C. Response to Reviewer #2

Comment 1

The predictive models still struggle with extreme outlier years, limiting robustness.

Response:

Acknowledged in Section 5 (Discussion) and the Conclusion.

Residual heavy-tailed patterns persist in extreme years; future research will refine jump-threshold calibration and employ event-based samples for enhanced robustness.

Comment 2

Reliance on annual panel data excludes high-frequency dynamics.

Response:

Addressed in the Conclusion.

We note that annual aggregation can smooth intra-year volatility and propose the use of quarterly or event-window data to better capture high-frequency wave–particle transitions.

Comment 3

Practical recommendations for firms and regulators are too broad and lack detailed guidance.

Response:

Expanded in the Conclusion, now offering concrete recommendations:

• Firms should monitor ROA–ROE divergence as an early-warning signal of structural imbalance;

• Regulators may track jump proportions as indicators of sectoral stress;

• Investors can incorporate jump risk alongside volatility in portfolio allocation.

Comment 4

The causal mechanisms behind the link between macroeconomic events and jump dynamics are not rigorously tested.

Response:

The Conclusion clarifies that current analyses are correlational and proposes future application of instrumental-variable and event-study designs to identify causal relationships.

Comment 5

Future research directions are presented vaguely without clear methodological pathways.

Response:

The Conclusion now outlines four explicit extensions:

(1) adoption of higher-frequency data,

(2) cross-industry validation,

(3) integration of behavioral-finance perspectives, and

(4) AI-driven interpretable modeling (e.g., contrastive GNNs, attention-based LSTMs).

Comment 6

Heavy reliance on physics analogies may risk conceptual oversimplification.

Response:

We sincerely appreciate this insightful comment.

In the revised manuscript, we have clarified that the “wave–particle duality” is not used as a literal physical analogy but as a conceptual framework that integrates continuous (wave-like) and discrete (particle-like) dynamics in corporate finance.

Specifically, in Section 2.3 (Interdisciplinary and Complex Systems Approaches) and the last paragraph of Section (Conclusion), we emphasize that this framework serves as a modeling metaphor rather than a physical parallel.

Its purpose is to unify two empirically observed states of financial behavior—

(1) gradual, continuous fluctuations (e.g., operational efficiency changes in ROA), and

(2) sudden, discrete jumps (e.g., policy or restructuring shocks affecting ROE).

This clarification reduces the risk of oversimplification and instead highlights the interdisciplinary advantage of using physical metaphors to describe complex adaptive financial systems.

The revised version thus presents the duality framework as an integrative modeling lens—consistent with Reviewer #2’s caution—rather than a literal application of quantum mechanics.

---

## [Editor Report · Decision Letter 1]

3 Nov 2025

The Wave-Particle Duality of Corporate Financial Metrics

PONE-D-25-40877R1

Dear Dr. ZHU,

We’re pleased to inform you that your manuscript has been judged scientifically suitable for publication and will be formally accepted for publication once it meets all outstanding technical requirements.

Kind regards,

Dipendra Karki, Ph.D.

Academic Editor

PLOS ONE

Additional Editor Comments (optional):

Dear Authors,

Thank you for submitting the revised version of your manuscript to PLOS ONE. I appreciate the time and effort you devoted to addressing the reviewers’ and editor’s comments with clarity and detail.

After carefully evaluating your revision and responses, I am pleased to accept your manuscript for publication in PLOS ONE.

Congratulations to you and your co-authors on this achievement, and thank you for choosing PLOS ONE as the venue for your research.

With best regards,

Dr. Dipendra Karki

Academic Editor

PLOS ONE
---

## [Editor Report · Acceptance letter]

PONE-D-25-40877R1

PLOS ONE

Dear Dr. Zhu,

I'm pleased to inform you that your manuscript has been deemed suitable for publication in PLOS ONE. Congratulations! Your manuscript is now being handed over to our production team.

Kind regards,

on behalf of

Dr. Dipendra Karki

Academic Editor

PLOS ONE